# Internet gaming disorder associated with depression among Thai medical students: A university-based cross-sectional study

Jarurin Pitanupong*, Jaruphan Sukhabote, Katti Sathaporn

Department of Psychiatry, Faculty of Medicine, Prince of Songkla University, Hat Yai, Songkhla, Thailand

* pjarurin@medicine.psu.ac.th

## Abstract

### Purpose

There is evidence that Internet Gaming Disorder (IGD) is strongly related to depression. Hence, this study aimed to investigate the prevalence of IGD among medical students, as well as to examine the association between IGD and depression.

### Participants and methods

From April to July 2024, a cross-sectional study surveyed Thai medical students studying in the first- to sixth-academic year. Participants were recruited using convenience sampling from one Faculty of Medicine and two Medical Education Centers located in southern Thailand. The survey utilized three questionnaires: Demographic and personal inquiry, the Nine-item Internet Gaming Disorder Scale (IGD Scale 9), and the Patient Health Questionnaire-9 (PHQ-9). Data were analyzed using descriptive statistics and logistic regression to assess the prevalence of IGD and its association with depression among the participants.

### Results

The survey of 634 medical students, 54.9% were female, with a median age of 20 years (IQR: 19–22). Notably, 8.4% reported IGD, while 21.6% exhibited symptoms of depression; indicated by a PHQ-9 score of 9 or higher. Of the 114 medical students who were both depressed and engaged in gaming, 26 (4.1%) reported IGD. Statistical analysis indicated significant differences between the IGD and non-IGD groups regarding gender ($p = 0.011$) and depression ($p < 0.001$). Furthermore, medical students experiencing depression were about 7.5 times more likely to have IGD (95% CI: 3.8–14.5), indicating a strong correlation between higher PHQ-9 scores and the severity of IGD.

**Data availability statement:** All relevant data are within the paper and its Supporting information files.

**Funding:** The author(s) received no specific funding for this work.

**Competing interests:** The authors have declared that no competing interests exist.

## Conclusion

This study emphasizes the complex relationship between IGD and mental health issues, such as depression, among medical students. Medical schools should establish early detection systems to identify these challenges. By prioritizing prevention strategies for both IGD and depression, medical schools can provide students with the tools to develop healthier coping mechanisms. Encouraging gaming alternatives can significantly enhance their overall well-being and mental health.

## Introduction

The Diagnostic and Statistical Manual of Mental Disorders, Fifth Edition, Text Revision (DSM-5-TR) defines Internet Gaming Disorder (IGD) as a pattern of excessive and prolonged participation in internet gaming, leading to cognitive and behavioral symptoms; such as loss of control, tolerance, and withdrawal [1]. In Thailand, the prevalence of IGD among students in a rural community school was 5.4% [2], while among university students, it was 14.3% [3]. A meta-analysis indicated that the pooled prevalence of IGD among medical students from various countries was 6.2%, approximately double the rate observed in the general population [4]. However, no prior studies have specifically examined the prevalence of IGD among Thai medical students, highlighting the need for this research.

Factors associated with IGD include: being male [5] and not living with both parents [6]. Maladaptive cognitions, achievement motivation, and rule-breaking behavior have been identified as risk factors [3]. Conversely, protective factors with IGD include self-control, a positive school climate, and engagement [7]. Recent studies indicate a strong correlation between the amount of time spent playing online games and the development of symptoms of depression. Specifically, playing for more than five hours a day increases the risk of developing depression [8,9].

IGD is strongly associated with depression [10]. A recent study indicated that various behavioral addictions; including IGD and Facebook addiction, are connected to symptoms of depression [11]. Individuals with IGD tend to experience more severe symptoms of depression and may continue to show symptoms of depression even during periods of remission from IGD [12]. Moreover, both IGD and depression share several overlapping symptoms; such as anhedonia, social withdrawal, fatigue, poor academic and work performance, and disrupted sleep-wake patterns [13–15]. This overlap emphasizes the need for early identification of both IGD and depression, particularly in vulnerable populations like medical students, where high stress and demands can exacerbate these issues [16].

A prior study reported the prevalence of depression among Thai medical students ranged from 11.1% to 15.9%, indicating a significant mental health concern [17,18]. Notably, no study has examined the prevalence of IGD within this demographic. Therefore, the primary objective of this study was to assess the prevalence of IGD among Thai medical students. The secondary objective was to explore the relationship between IGD and depression. Understanding the associations between IGD and

depression can help in implementing integrated treatment approaches, particularly prevention programs, that may mitigate their impact on mental health and academic success.

## Materials and methods

### Respondents and procedure

After receiving approval from the ethics committees at the Faculty of Medicine, Prince of Songkla University (REC: 67-033-3-4), this cross-sectional study was conducted among all first- to sixth-academic-year medical students studying at one Faculty of Medicine, and two Medical Education Centers, in southern Thailand. In the context of medical education in Thailand, the curriculum is divided into two phases. The first three years, known as the pre-clinical phase, focus on acquiring basic scientific knowledge in subjects like anatomy, physiology, and biochemistry. The final three years, referred to as the clinical phase, shift the focus towards practical application. Students in this phase engage in both theoretical coursework and hands-on patient care under the supervision of experienced clinicians.

From April to July 2024, medical students were surveyed online. Participants must have been at least 18 years old, fluent in Thai, and willing to complete all questionnaires. Those unable to finish the questionnaire or purposefully withdrew from the study were excluded from the analysis.

Data collection for this study was conducted using a convenience sampling method. The researcher utilized personal contacts to obtain permission and provide information about the study to the administrators of the official social media accounts used by medical students. This approach ensured that the sample consisted exclusively of medical student members, facilitating targeted recruitment while maintaining relevance to the study's focus. Since the participants in this study completed the questionnaires anonymously, they first read the participant's information sheet and agreed to the terms by clicking "Agree and Proceed," which led them to the questionnaires. By joining the questionnaires, participants implied their consent to participate in the research, eliminating the need for a signed informed consent form. The requirement for a signature was waived by the Medical Ethics Committee of the Faculty of Medicine, Prince of Songkla University.

Participants were invited to join the study by clicking a provided link or scanning a QR code shared through social media advertisements to access the online questionnaire. They had the flexibility to complete and submit the questionnaires immediately or later. Participants retained the right to withdraw from the study at any point without needing to provide a reason. To ensure confidentiality, signatures were not required from respondents, and all data were securely stored. Access to the information was limited to the researcher, with the data protected by a password. This approach safeguarded participants' privacy while allowing for the collection of valuable data for the study.

The sample calculation used the prevalence of IGD among medical students at 6.2% [4]. The researcher used sample size calculation for estimating proportion, with a margin of error of 2% and a 95% confidence level, resulting in the need for a sample size of at least 559 subjects.

### Questionnaires

I Demographic and personal information included: age, gender, religion, academic year, cumulative Grade Point Average (GPA), physical and psychiatric illnesses, history of alcohol and substance use, and current life stress.

II The Nine-Item Internet Gaming Disorder Scale (IGD Scale 9)-Thai version is a self-rating questionnaire designed to evaluate IGD, comprising nine questions, based on the DSM-5 criteria. Responses are yes/no, with "yes" scoring one point and "no" scoring zero points, resulting in a total score ranging from 0 to 9. A cut-off score of 5 indicates the criteria for IGD. The IGD-9 scale has demonstrated a Cronbach's alpha coefficient of 0.69, with a sensitivity of 0.69 and specificity of 0.91 [19].

III The Patient Health Questionnaire-9 (PHQ-9)- Thai version is widely recognized as an effective tool for screening and assessing the severity of depression. It consists of a 4-point scale for each of the nine questions: 0 (never), 1 (rarely), 2

(sometimes), and 3 (always). Total scores range from 0 to 27, with interpretations as follows: 0−4 (no or minimal depression), 5−9 (mild depression), 10−14 (moderate depression), 15−19 (moderately severe depression), and 20−27 (severe depression) [20]. A cut-off score of nine or higher is considered indicative of depression. In terms of psychometric properties, the PHQ-9 demonstrated internal consistency with a Cronbach's alpha coefficient of 0.79, along with a sensitivity of 0.53 and a specificity of 0.98 [21].

### Statistical analysis

Descriptive statistics were calculated using proportions, means, standard deviations (SD), medians, and interquartile ranges (IQR). Data distribution was analyzed using the Shapiro-Wilk test. Group comparisons were performed using Fisher's exact test, Wilcoxon rank-sum test, and Chi-square test. Logistic regression analysis was conducted using R 4.3.1 Software (The R Foundation for Statistical Computing, Vienna, Austria), with odds ratio (OR) and 95% confidence interval (CI) to estimate the effect sizes. A $p$-value of < 0.05 was considered to indicate significant statistical differences.

## Results

### Demographic characteristics

From April to July 2024, 967 medical students were surveyed, with 634 agreeing to participate and complete the questionnaires. No significant differences were found between the collaborative and non-collaborative groups of medical students regarding key demographic factors, such as gender and year of education.

The participants were predominantly female (54.9%), Buddhist (85.3%), and in their preclinical years (61.0%), with a median age of 20.0 years (IQR:19.0–22.0). Most participants reported no history of physical illness (85.8%) or psychiatric illness (93.4%). Among those with physical illnesses, the most common conditions reported were allergies (8.0%), thyroid disease (0.8%), asthma (0.6%), dyspepsia (0.3%), and migraines (0.3%). Regarding psychiatric conditions, the most frequently reported were major depressive disorder (3.2%), generalized anxiety disorder (1.7%), adjustment disorder (0.8%), and attention deficit hyperactivity disorder (0.6%). Less than one-third of the participants (30.8%) reported a history of alcohol use, primarily involving social drinking occurring 1–2 times per month in 7.9% of cases. Nearly half of the medical students (49.8%) reported experiencing current life stress, primarily due to academic and examination-related pressures (37.1%) (Table 1).

### Internet gaming disorder

The survey of 634 medical students revealed that 510 medical students (80.4%) had engaged in gaming within the past year. Among these gamers, the median score on the IGD-9 scale was 1.0, with an IQR of 1.0 to 3.0. Notably, only 43 medical students (8.4%) were classified as having IGD, based on an IGD-9 score ranging from 5 to 9.

### Depression

In the survey of 634 medical students, the median PHQ-9 score was 5.0, with an IQR of 2.0 to 8.0. Notably, 137 medical students (21.6%) scored nine or higher on the PHQ-9, indicating the presence of depression [21]. Regarding the level of depression, medical students reported experiencing moderate to severe depression, as shown in Fig 1.

The main symptoms reported among the 137 medical students indicating depression included loss of interest in activities (69.9%), difficulty concentrating (60.3%), fatigue (59.8%), and sleep disturbances (54.1%), as shown in Fig 2.

Additionally, of 114 medical students (17.9%) who were both depressed and engaged in gaming within the past year, 26 students (4.1%) were classified as having IGD. However, no statistically significant differences were found in the main symptoms of depression between the IGD and non-IGD groups ($p$-value > 0.05, range from 0.164 to 1) (Fig 3).

**Table 1. Demographic characteristics (N = 634).**

| Demographic characteristics | Number (%) |
|---|---|
| **Gender** | |
| Male | 244 (38.5) |
| Female | 348 (54.9) |
| LGBTQ+ | 40 (6.3) |
| No answer | 2 (0.3) |
| **Year of education** | |
| Pre-clinic | 387 (61.0) |
| Clinic | 247 (39.0) |
| **Religion** | |
| Buddhism | 541 (85.3) |
| Islam, Christianity, others | 93 (14.7) |
| **Physical illness** | |
| No | 544 (85.8) |
| Yes | 90 (14.2) |
| **Psychiatric illness** | |
| No | 592 (93.4) |
| Yes | 42 (6.6) |
| **History of alcohol use** | |
| No | 439 (69.2) |
| Yes | 195 (30.8) |
| **History of substance use** | |
| No | 628 (99.1) |
| Yes (Nicotine, E-cigarettes) | 6 (0.9) |
| **Presence of current life stress** | |
| No | 318 (50.2) |
| Yes | 316 (49.8) |
| **Game playing in the past 12 months** | |
| No | 124 (19.6) |
| Yes | 510 (80.4) |

### Association between demographic characteristics, depression, and internet gaming disorder

From an initial sample of 634 medical students, the study examined the association between demographic characteristics, depression, and IGD among 510 students who had engaged in gaming within the past year. Variables with a *p*-value less than 0.2 from the bivariate analysis, along with the main variable of interest (depression) presented in Table 2, were considered potential candidates for inclusion in the multivariate logistic regression model. The candidate variables included in the initial model for multivariate analysis were gender, presence of physical illness, history of substance use, current stress, and depression. Multicollinearity among these variables was assessed using the Variance Inflation Factor (VIF). Due to only two non-responses for gender, these were excluded from both the bivariate and multivariate analyses. Multivariable logistic regression with stepwise backward elimination was used. In the final model, variables with a *p*-value less than 0.05 were considered statistically significant and associated with the outcome. The results showed a statistically significant difference between the depression and non-depression groups regarding the presence of IGD. After controlling for gender, medical students with depression were about 7.5 times more likely to have IGD compared to those without

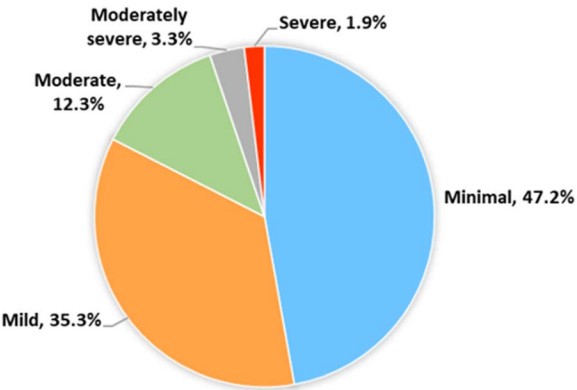

**Fig 1. Level of depression (N = 634).**

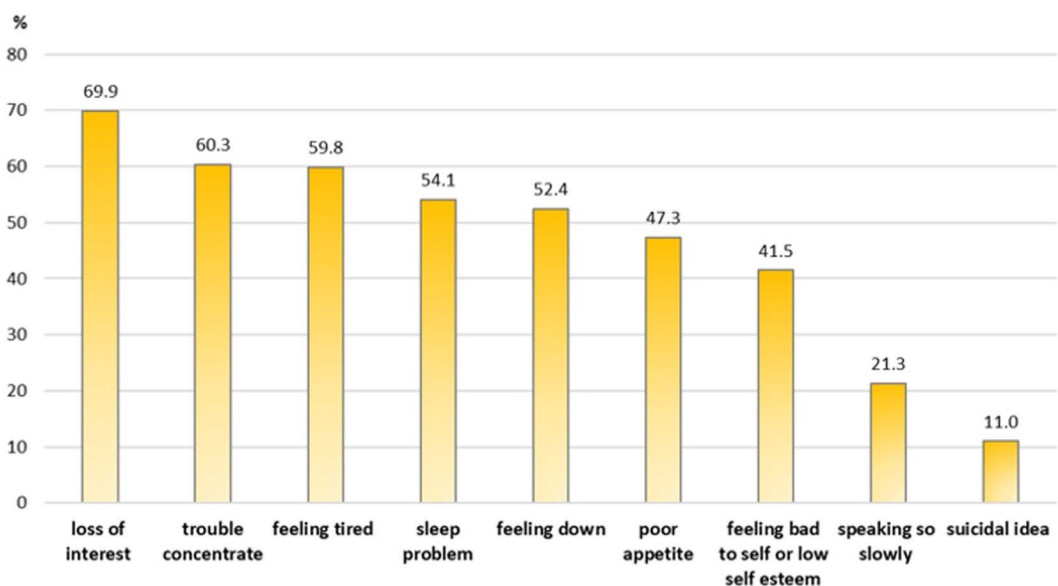

**Fig 2. Frequency of symptoms reported among medical students with depression (N = 137).**

depression (95% CI: 3.8–14.5). This suggests a strong association between higher PHQ scores and more severe IGD, indicating that increased depression levels are related to a greater likelihood of experiencing IGD (Table 3).

## Discussion

This study provided valuable information on the prevalence of IGD among medical students and its association with depression. The finding that 8.4% of medical students were classified as having IGD was relatively lower than the 14.3% prevalence found among university students in Thailand [3]; however, it was consistent with global trends in medical students [4]. The lower prevalence of IGD among medical students compared to the general university student population might reflect the unique stressors and demands of medical education. Medical students tend to have a more rigorous academic schedule and often face higher levels of stress. This may leave less time for gaming, or they may prioritize other activities that better align with their career goals [16]. This can contribute to a lower rate of IGD within this group.

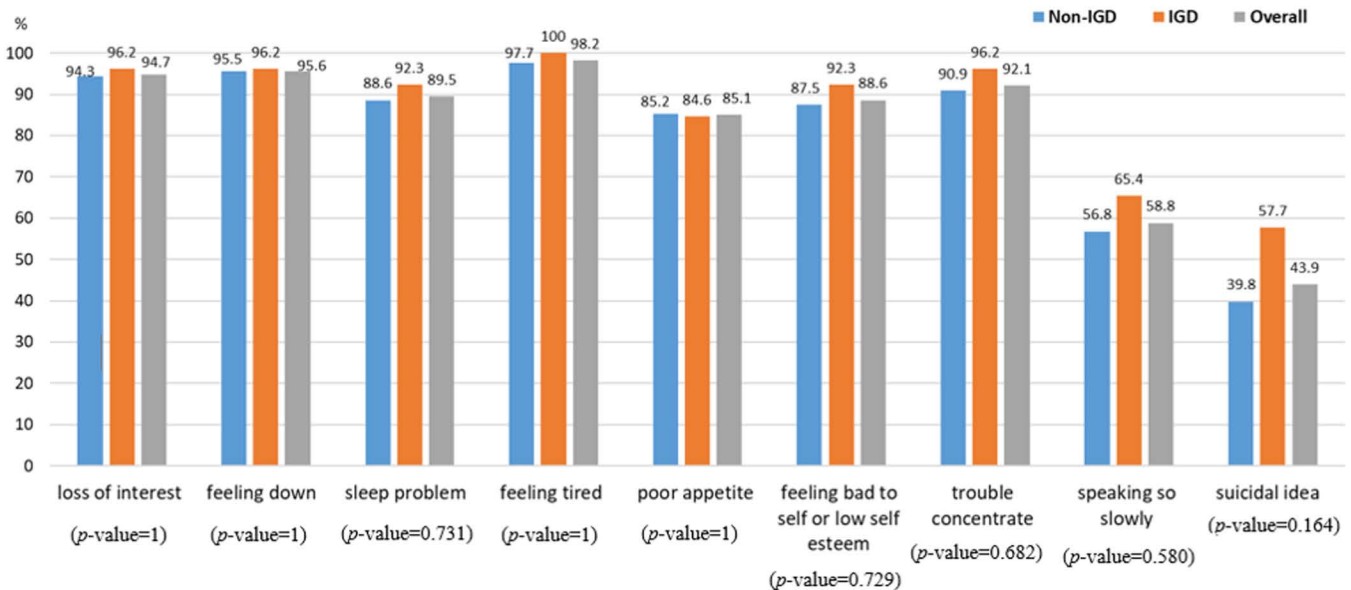

**Fig 3. Frequency of symptoms of depression among medical students with internet gaming disorder (N = 26) and those without internet gaming disorder (N = 88).**

The observed significant differences between the IGD and non-IGD groups, in terms of gender and depression, in this study align with findings from prior studies. This provides additional insights into the gender-specific patterns of gaming behavior and its connection with mental health. Males are more vulnerable to developing IGD than females because males are generally more likely to engage in excessive gaming, which may be linked to both social and biological factors. For instance; gaming culture often appeals more to males, and social norms may encourage higher levels of gaming among men [22]. Moreover, the prior study found that males exhibit greater activation in brain regions; such as the medial frontal gyrus, bilateral middle temporal gyri, and the thalamus, compared to females during gaming. These brain regions are associated with reward processing, and decision-making, which are critical in gaming behavior [23].

The findings of this study, particularly the high rate of depression (21.6%) among medical students and the strong association between depression and IGD underline a critical mental health concern in this population. The fact that medical students with depression were 7.5 times more likely to also suffer from IGD underscores the importance of addressing both conditions concurrently. If both depression and IGD are treated together, it could lead to better outcomes for medical students' mental health and overall well-being. The prior study suggests that effective pharmacological treatments for depression could also alleviate symptoms of IGD [24]. Moreover, integrated treatment (for both IGD and comorbid depression) has led to reductions in IGD symptoms; ranging from 15.4% to 51.4% [25–27]. This suggests that by focusing on concurrent treatment for both depression and IGD, medical schools can help medical students develop healthier coping mechanisms, reduce the potential for academic and personal distress, and enhance overall well-being. Early identification, accessible treatment, and comprehensive support programs can have a significant positive impact, helping medical students thrive both academically and emotionally [16]. Moreover, creating a safe and supportive medical school learning environment, along with fostering good relationships between medical students and faculty members, is crucial. Ensuring open communication, promoting a culture of empathy, and encouraging understanding within medical schools can significantly contribute to the well-being of students. This approach can enhance their overall academic experience and resilience in the face of challenges, ultimately helping to reduce mental health problems among medical students [17,28].

**Table 2. Bivariate analysis of the association between demographic characteristics and having the presence of internet gaming disorder (N = 510).**

| Demographic characteristics | IGD | | Chi2 P-value |
|---|---|---|---|
| | **Yes (N = 43)** | **No (N = 467)** | |
| **Gender** | | | 0.185 |
| Male | 21 (48.8) | 196 (42.2) | |
| Female | 22 (51.2) | 237 (51.0) | |
| LGBTQ+ | 0 (0.0) | 32 (6.9) | |
| **Age (years)** | | | 0.479[a] |
| Median (IQR) | 20 (19, 22) | 20 (19, 22) | |
| **Grade Point Average (GPA)** | | | 0.716[a] |
| Median (IQR) | 3.4 (3.1, 3.6) | 3.4 (3.1, 3.6) | |
| **Year of education** | | | 0.231 |
| Pre-clinic | 24 (55.8) | 309 (66.2) | |
| Clinic | 19 (44.2) | 158 (33.8) | |
| **Religion** | | | 1 |
| Buddhism | 37 (86.0) | 405 (86.7) | |
| Islam, Christianity, others | 6 (14.0) | 62 (13.3) | |
| **Physical illness** | | | 0.074 |
| No | 32 (74.4) | 401 (85.9) | |
| Yes | 11 (25.6) | 66 (14.1) | |
| **Psychiatric illness** | | | 0.732[b] |
| No | 40 (93.0) | 440 (94.2) | |
| Yes | 3 (7.0) | 27 (5.8) | |
| **History of alcohol use** | | | 0.483 |
| No | 31 (72.1) | 306 (65.5) | |
| Yes | 12 (27.9) | 161 (34.5) | |
| **History of substance use** | | | 0.084[b] |
| No | 41 (95.3) | 463 (99.1) | |
| Yes (Nicotine, E-cigarettes) | 2 (4.7) | 4 (0.9) | |
| **Presence of current life stress** | | | 0.009 |
| No | 13 (30.2) | 244 (52.2) | |
| Yes | 30 (69.8) | 223 (47.8) | |
| **Depression** | | | < 0.001 |
| No (PHQ-9 score < 9) | 17 (39.5) | 379 (81.2) | |
| Yes (PHQ-9 score ≥ 9) | 26 (60.5) | 88 (18.8) | |

**Note:** a = Wilcoxon rank-sum test, b = Fisher's exact test, IQR = interquartile ranges,

IGD= Internet Gaming Disorder, PHQ-9= The Patient Health Questionnaire-9.

Despite the association, the finding that there were no significant differences in symptoms of depression between medical students with and without IGD suggests that IGD may coexist with depression without necessarily altering the core symptomatology of depression [13]. This indicates the complexity of how mental health and gaming behavior intersect. Therefore, further study is necessary to explore the unique factors influencing IGD among medical students and to investigate how IGD and depression interact [29].

**Table 3. Factor associated with internet gaming disorder (N = 508).**

| Factors | Crude odds ratio (95% CI) | Adjusted odds ratio (95% CI) | *P*-value LR-test |
|---|---|---|---|
| **Gender** | | | 0.011 |
| Female | Reference | Reference | |
| Male | 1.2 (0.6, 2.2) | 1.4 (0.7, 2.7) | |
| LGBTQ+ | 0 (0, Inf) | 0 (0, Inf) | |
| **Depression** | | | < 0.001 |
| No (PHQ-9 score < 9) | Reference | Reference | |
| Yes (PHQ-9 score ≥ 9) | 6.7 (3.5, 12.8) | 7.5 (3.8, 14.5) | |

**Note:** There was small sample size for LGBTQ+, LR-test = likelihood ratio.

This study highlights the importance of integrated treatment approaches in medical schools to address IGD and depression through targeted mental health programs effectively. While it is the first investigation of its issue in Southern Thailand in the past decade, it has notable limitations. This study is a cross-sectional design that limits the ability to infer causality or long-term trends. For instance; while the relationship between IGD and depression is highlighted, it is difficult to determine which comes first or how these conditions might influence each other over time. Moreover, relying on self-administered questionnaires can introduce biases, as participants may underreport or over-report symptoms based on various factors; such as social desirability or misunderstanding of the questions. This could make the results less reliable. In addition, there is the potential for sample bias, with medical students experiencing high levels of depression or IGD either avoiding participation or being overly represented due to a heightened interest in the survey. This could distort the findings, leading to an inaccurate estimate of the true prevalence of these conditions. As the study is focused on Southern Thailand, it may not be fully representative of medical students across the entire country. Cultural and regional differences could influence both the prevalence and the experience of depression and IGD, so generalizing the findings to other regions may not be entirely appropriate without further studies. A previous study found a strong correlation between the amount of time spent playing online games and the development of depressive symptoms. In particular, playing for more than five hours a day was associated with an increased risk of developing depression [8]. However, this study did not explore the characteristics or patterns of gameplay, which limits the depth of its findings and their applicability in creating specific guidelines for the prevention of IGD.

Future studies could provide more detailed insights into gaming behaviors, including the types of games played, the frequency and duration of gaming sessions, and the psychological factors that influence these habits. Additionally, adopting a longitudinal design would allow for a better understanding of the long-term effects of gaming and how it may contribute to the development of IGD. Moreover, employing more diverse sampling methods and including multiple regions in Thailand would help capture a broader range of experiences, accounting for cultural and regional differences that could influence gaming behavior. These improvements would contribute to creating a clearer and more generalizable understanding of how depression and IGD affect medical students across the country, as well as inform more effective mental health interventions tailored to the specific needs of this population.

## Conclusion

Nearly one-tenth of medical students reported experiencing IGD, with a significant relation identified between male gender, depression, and IGD. These findings emphasize the importance of addressing both depression and IGD within medical schools, as targeting these issues could enhance medical students' mental health and overall well-being. Implementing support systems and targeted interventions could be beneficial in alleviating the effects of these disorders. By

fostering a proactive approach to mental health, medical schools can provide students with the tools to develop healthier coping mechanisms. Encouraging gaming alternatives can significantly enhance their well-being and mental health.

## Acknowledgments

The authors would like to acknowledge the participants for their willingness to offer information. We would like to also acknowledge the team's research assistants; Professor Hutcha Sriplung, Nisan Werachattawan, and Kruewan Jongbor-wanwiwat, for their support. The English of this article was proofread/edited by the Office of International Affairs, Faculty of Medicine, Prince of Songkla University.

## Author contributions

**Conceptualization:** Jarurin Pitanupong, Jaruphan Sukhabote, Katti Sathaporn.

**Data curation:** Jarurin Pitanupong, Jaruphan Sukhabote, Katti Sathaporn.

**Formal analysis:** Jarurin Pitanupong, Jaruphan Sukhabote, Katti Sathaporn.

**Investigation:** Jarurin Pitanupong, Jaruphan Sukhabote, Katti Sathaporn.

**Methodology:** Jarurin Pitanupong.

**Project administration:** Jaruphan Sukhabote.

**Supervision:** Jarurin Pitanupong.

**Validation:** Jarurin Pitanupong.

**Visualization:** Jarurin Pitanupong.

**Writing – original draft:** Jarurin Pitanupong, Jaruphan Sukhabote, Katti Sathaporn.

**Writing – review & editing:** Jarurin Pitanupong.

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
