## [Decision Letter · Decision Letter 0]

15 Jan 2025

Dear Dr. Pitanupong,

Thank you for submitting your manuscript to PLOS ONE. After careful consideration, we feel that it has merit but does not fully meet PLOS ONE’s publication criteria as it currently stands. Therefore, we invite you to submit a revised version of the manuscript that addresses the points raised during the review process.

We look forward to receiving your revised manuscript.

Kind regards,

Daniel Ahorsu, PhD

Academic Editor

PLOS ONE

Journal Requirements:

3. In this instance it seems there may be acceptable restrictions in place that prevent the public sharing of your minimal data. However, in line with our goal of ensuring long-term data availability to all interested researchers, PLOS’ Data Policy states that authors cannot be the sole named individuals responsible for ensuring data access (http://journals.plos.org/plosone/s/data-availability#loc-acceptable-data-sharing-methods ).

4.Your ethics statement should only appear in the Methods section of your manuscript. If your ethics statement is written in any section besides the Methods, please delete it from any other section.

Reviewers' comments:

Reviewer's Responses to Questions

**Comments to the Author**

1. Is the manuscript technically sound, and do the data support the conclusions?

Reviewer #1: Yes

Reviewer #2: Yes

2. Has the statistical analysis been performed appropriately and rigorously?

Reviewer #1: Yes

Reviewer #2: No

3. Have the authors made all data underlying the findings in their manuscript fully available?

Reviewer #1: Yes

Reviewer #2: No

4. Is the manuscript presented in an intelligible fashion and written in standard English?

Reviewer #1: No

Reviewer #2: Yes

Reviewer #1: Good topic to discuss, but needs some clarification

1- what types of physical illness , and could that affects results

2- you didn't discuss the frequency of Internet gaming , just yes or no , though this may affect results

3- alcohol and smoking must be clarified more like duration and frequency per day

4- manuscript needs English proof

Reviewer #2: Review comments

General comments.

Any study that involves medical students is always of interest especially those that may affect their practice as clinicians in less than no time. The Authors are commended for conceptualizing the study and for their efforts in ensuring that their research work is published. I am of the opinion that the manuscript has merit but may require revision before it could be accepted for publication.

Authors should take note of the following observations:

1. Title of study: It appears as if Authors were out to establish a relationship between Internet Gaming Disorder (IGD) and symptoms of depression. This was very evident in the title of the study, the approach to statistical analysis and in the recommendations. If this approach should stand, the inclusion of a conceptual framework will be very essential.

2. Conclusion and Recommendation: Authors recommended the concurrent treatment for both IGD and depression and development of healthier coping mechanisms. This recommendation is rather far reaching. In a study involving future clinicians, the focus ought to have been preventive. Thus measures like how to improve the mental health of the students, ensure good interactions between the students and the Faculty members and the early identification of symptoms of depression and IGD ought to have been of priority. Authors should also indicate the implications of study findings.

3. Sample size determination: A sample size formula was applied in determining the sample size. Authors went ahead to indicate a response rate. This (the response rate) may not be necessary since this was not a total population study.

4. Sampling technique: Authors indicated that a convenience sampling approach was employed in the study. An online questionnaire was used. How did the Authors ensure that the exclusion criteria were adhered to?

5. The study limitation should include the method used in recruiting respondents for the study. The results may not been a true representation of the issues at stake based on the sampling technique.

6. Study instrument: Authors should provide a reference for the questionnaires. A reference should also be provided for the cut off of 9 used for Patient Health Questionnaire. What does the Authors mean that, ‘For this study, the IGD scale 9 exhibited greater internal consistency with Cronbach’s alpha coefficient of 0.71.’ Same comment was made for Patient Health Questionnaire.

7. Medical education in Thailand: A brief description of the training of medical students in Thailand will be helpful. For example in table 1, categories like ‘Pre-clinic’ and ‘Clinic’ were indicated. What do they mean? How many years are spent in Thailand in studying Medicine?

8. This statement should be clarified; ‘The finding that there was no significant difference in symptoms of depression between medical students with or without IGD….’ Which table showed this result?

9. Statistical analysis: The age of the students is expected to be normally distributed and should be represented using mean and standard deviation since it is a continuous variable. For a study that included a binary logistic regression analysis, adjusted odds ratio and 95% confidence interval ought to have been used to show the predictors of the outcome measure of the study instead of p value in the abstract. Why was table 2 and 3 separated? Authors indicated that a p value of <0.2 was used in including variables into the logistic regression model after bivariate analysis; was this adhered to? The column for IGD, ‘Yes’ should come first in table 2. How were the two non-responses for gender taken care in bivariate and multivariate analysis in tables 2 and 3? This should be explained? A proportion of 30.8% should be reported as less than one third. Authors should also include a correlation analysis between IGD and depression scores.

10. Comments related to studies ought to have more than one reference. Authors should avoid the use of Arabic numerals in the body of the manuscript. This may conflict with the reference numbers. Include Thailand as a key word. Instead of signs of depression, use symptoms of depression. In line 75, review the use of ‘..rises to 14.3%’.

11. Questionnaire 1 as indicated by the Authors included the CGPA. This was not included in the result section. This should be clarified. If there was a variable like ‘Perceived academic performance’ it could have also been useful to see its relationship with IGD and symptoms of depression. The categories for depression in tables 2 and 3 should be renamed. Depression symptoms, ‘Yes’ and ‘No’ will be more appropriate.

12. Authors should review the manuscript critically for minor errors of grammar.

**Do you want your identity to be public for this peer review?** For information about this choice, including consent withdrawal, please see our Privacy Policy

Reviewer #1: No

Reviewer #2: **Yes: ** EDMUND NDUDI OSSAI

---

## [Author Response · Author response to Decision Letter 1]

17 Feb 2025

Thanks to your suggestions, our manuscript has become more scientifically valuable and elegant. We sincerely hope to receive your kindness again and forever.

---

## [Decision Letter · Decision Letter 1]

6 Apr 2025

Dear Dr. Pitanupong,

Thank you for submitting your manuscript to PLOS ONE. After careful consideration, we feel that it has merit but does not fully meet PLOS ONE’s publication criteria as it currently stands. Therefore, we invite you to submit a revised version of the manuscript that addresses the points raised during the review process.

We look forward to receiving your revised manuscript.

Kind regards,

Dahua Yu

Academic Editor

PLOS ONE

Journal Requirements:

Reviewers' comments:

Reviewer's Responses to Questions

**Comments to the Author**

Reviewer #2: (No Response)

Reviewer #3: All comments have been addressed

2. Is the manuscript technically sound, and do the data support the conclusions?

Reviewer #2: Yes

Reviewer #3: Yes

3. Has the statistical analysis been performed appropriately and rigorously?

Reviewer #2: No

Reviewer #3: Yes

4. Have the authors made all data underlying the findings in their manuscript fully available?

Reviewer #2: Yes

Reviewer #3: Yes

5. Is the manuscript presented in an intelligible fashion and written in standard English?

Reviewer #2: Yes

Reviewer #3: Yes

Reviewer #2: Review comments

General comments

I commend the Authors for the good review of the manuscript. I am also of the opinion that certain aspects of the manuscript may require another revision before the manuscript could be accepted for publication.

Authors should take note of the following observations:

1. Journal guidelines. Authors should adhere to Journal guidelines in presenting the manuscript including the abstract. Authors should refer to the website of the Journal.

2. Abstract. Authors should delete the Arabic numerals used in listing the number of questionnaires used in the study. Do the same in the body of the manuscript. In the conclusion, Authors should focus on preventive measures instead of treatment options. Some of the results in the abstract were not included in the results section. Thailand should be included as key words.

3. Statistical analysis. In line 213, Authors stated that ‘variables with a p value of <0.2 from the univariate analysis were selected for inclusion …….’. Authors should change univariate to bivariate. In line 222, Authors stated that ‘we incorporated significant variables into the multivariate logistic regression model…..’. The two statements are not in tandem. I align myself with the first statement as corrected. Why were ‘physical illness’ ‘history of substance use’ and ‘presence of current life stress’ not included in the logistic regression model? A revision is required.

4. Conceptual framework. I am of the opinion that based on the title of the study, a conceptual framework may be required to solidify the study concept. Adhering to comment 3 above may establish the relevance of the title and not that the analysis was based on being selective.

5. Introduction. Change ‘rises’ in line 75. A meta-analysis could not have been done on medical students. Line 85, should have more than one reference since it was based on recent studies. Line 104, Were more than one ethics committee involved in approving the study?

6. All acronyms used in tables should be explained as foot notes

Reviewer #3: (No Response)

**Do you want your identity to be public for this peer review?** For information about this choice, including consent withdrawal, please see our Privacy Policy

Reviewer #2: **Yes: ** EDMUND NDUDI OSSAI

Reviewer #3: No

---

## [Author Response · Author response to Decision Letter 2]

8 Apr 2025

Thanks so much for everythig that support our team.

---

## [Decision Letter · Decision Letter 2]

18 May 2025

Dear Dr. Pitanupong,

Thank you for submitting your manuscript to PLOS ONE. After careful consideration, we feel that it has merit but does not fully meet PLOS ONE’s publication criteria as it currently stands. Therefore, we invite you to submit a revised version of the manuscript that addresses the points raised during the review process.

plosone@plos.org. A rebuttal letter that responds to each point raised by the academic editor and reviewer(s). You should upload this letter as a separate file labeled 'Response to Reviewers'.A marked-up copy of your manuscript that highlights changes made to the original version. You should upload this as a separate file labeled 'Revised Manuscript with Track Changes'.An unmarked version of your revised paper without tracked changes. You should upload this as a separate file labeled 'Manuscript'.

We look forward to receiving your revised manuscript.

Kind regards,

Dahua Yu

Academic Editor

PLOS ONE

Journal Requirements:

Reviewers' comments:

Reviewer's Responses to Questions

**Comments to the Author**

Reviewer #2: (No Response)

2. Is the manuscript technically sound, and do the data support the conclusions?

Reviewer #2: Yes

3. Has the statistical analysis been performed appropriately and rigorously?

Reviewer #2: No

4. Have the authors made all data underlying the findings in their manuscript fully available?

Reviewer #2: Yes

5. Is the manuscript presented in an intelligible fashion and written in standard English?

Reviewer #2: Yes

Reviewer #2: Review comments

I remain grateful to the Authors for the good review of the manuscript. My intention was that the last review should be the last revision for the manuscript signaling the recommendation for the acceptance of the manuscript for publication.

However, a careful review of the manuscript indicated that there are some contradictions in the manuscript which should be reconciled before the manuscript could be accepted for publication.

1. In lines 147 – 149 of the manuscript indicated how the total depressive score of respondents should be categorized. “Total scores range from 0 to 27, with interpretations as follows: 0-4 (no or minimal depression), 5-9 (mild depression), 10-14 (moderate depression), 15-19 (moderately severe depression), and 20-27 (severe depression).”

Commencing from the same line 149 Authors stated that ‘a cut off score of nine or higher is considered indicative of depression.’ No reference was provided for the use of cut off score of 9. This should be clarified.

2. The report in table is different from that in table 2 with regards to the sample size of the study. In table 1, the total number of respondents was 634. This was indicated in line 163. The total number as indicated in table 2 is 510 except gender. This should be reconciled.

3. Authors should refer to lines 206 -210. However, in line 217, Authors indicated the use of significant variables as basis for inclusion into the logistic regression model after bivariate analysis. These comments as indicated make table 3 very confusing. Authors should state the basis for inclusion of variables into the regression model after applying the Chi square test.

4. Authors should include convenience sampling in the abstract.

5. Authors should use Roman numerals instead of Arabic in list the study instrument in the body of the manuscript. (Lines 135, 138 and 144).

**Do you want your identity to be public for this peer review?** For information about this choice, including consent withdrawal, please see our Privacy Policy

Reviewer #2: **Yes: ** EDMUND NDUDI OSSAI

---

## [Author Response · Author response to Decision Letter 3]

19 May 2025

Thanks for your kindness and patient. We wait and hope for your positive response, Thanks so much.

---

## [Decision Letter · Decision Letter 3]

23 June 2025

Internet gaming disorder associated with depression among Thai medical students: a university-based cross-sectional study

PONE-D-24-54168R3

Dear Dr. Pitanupong,

We’re pleased to inform you that your manuscript has been judged scientifically suitable for publication and will be formally accepted for publication once it meets all outstanding technical requirements.

Kind regards,

Dahua Yu

Academic Editor

PLOS ONE

Additional Editor Comments (optional):

Reviewers' comments:

Reviewer's Responses to Questions

**Comments to the Author**

Reviewer #2: All comments have been addressed

2. Is the manuscript technically sound, and do the data support the conclusions?

Reviewer #2: Yes

3. Has the statistical analysis been performed appropriately and rigorously?

Reviewer #2: Yes

4. Have the authors made all data underlying the findings in their manuscript fully available?

Reviewer #2: Yes

5. Is the manuscript presented in an intelligible fashion and written in standard English?

Reviewer #2: Yes

Reviewer #2: Review comments

General comments

I commend the efforts of the Authors for the good review of this manuscript. I am of the opinion that the manuscript should be accepted for publication.

**Do you want your identity to be public for this peer review?** For information about this choice, including consent withdrawal, please see our Privacy Policy

Reviewer #2: **Yes: ** EDMUND NDUDI OSSAI

---

## [Editor Report · Acceptance letter]

PONE-D-24-54168R3

PLOS ONE

Dear Dr. Pitanupong,

I'm pleased to inform you that your manuscript has been deemed suitable for publication in PLOS ONE. Congratulations! Your manuscript is now being handed over to our production team.

Kind regards,

on behalf of

Prof. Dahua Yu

Academic Editor

PLOS ONE